# Unveiling the dynamic active site of defective carbon-based electrocatalysts for hydrogen peroxide production

Qilong Wu[1,2,3,11], Haiyuan Zou[4,11], Xin Mao[5,11], Jinghan He[1], Yanmei Shi[6], Shuangming Chen[7], Xuecheng Yan[3], Liyun Wu[1], Chengguang Lang[3,8], Bin Zhang[6], Li Song[7], Xin Wang[9,10], Aijun Du[5], Qin Li[3], Yi Jia[9,10] ✉, Jun Chen[2] ✉ & Xiangdong Yao[1,8] ✉

Active sites identification in metal-free carbon materials is crucial for developing practical electrocatalysts, but resolving precise configuration of active site remains a challenge because of the elusive dynamic structural evolution process during reactions. Here, we reveal the dynamic active site identification process of oxygen modified defective graphene. First, the defect density and types of oxygen groups were precisely manipulated on graphene, combined with electrocatalytic performance evaluation, revealing a previously overlooked positive correlation relationship between the defect density and the 2 e⁻ oxygen reduction performance. An electrocatalytic-driven oxygen groups redistribution phenomenon was observed, which narrows the scope of potential configurations of the active site. The dynamic evolution processes are monitored via multiple in-situ technologies and theoretical spectra simulations, resolving the configuration of major active sites (carbonyl on pentagon defect) and key intermediates (*OOH), in-depth understanding the catalytic mechanism and providing a research paradigm for metal-free carbon materials.

Currently, the most widely used chemical fuels, including hydrogen, ammonia, and hydrogen peroxide, are generally produced and refined via a series of large-scale centralized reactors[1,2]. In a typical industrial synthetic route for hydrogen peroxide ($H_2O_2$), the anthraquinone oxidation/reduction process, involves energy-intensive and large-scale purification and refinement steps[3,4]. There is a general call for the development of innovative synthetic prototype where both $H_2O_2$ synthesis and storage should be closer to the point of consumption,

[1]State Key Laboratory of Inorganic Synthesis and Preparative Chemistry, College of Chemistry, Jilin University, Changchun 130012, PR China. [2]Intelligent Polymer Research Institute, Australian Institute for Innovative Materials, Innovation Campus, University of Wollongong, Squires Way, North Wollongong, NSW 2500, Australia. [3]School of Environmental engineering and Built Environment, Griffith University, Nathan Campus, Brisbane, QLD 4111, Australia. [4]Guangdong Provincial Key Laboratory of Energy Materials for Electric Power, Southern University of Science and Technology, Shenzhen 518055, China. [5]School of Chemistry and Physics and Centre for Materials Science, Queensland University of Technology, Gardens Point Campus, Brisbane, QLD 4001, Australia. [6]School of Science, Institute of Molecular Plus, Tianjin University, Tianjin 300072, China. [7]Hefei National Laboratory for Physical Sciences at the Microscale, iChEM (Collaborative Innovation Center of Chemistry for Energy Materials), School of Chemistry and Materials Science, and National Synchrotron Radiation Laboratory, University of Science and Technology of China, Hefei, Anhui 230026, PR China. [8]School of Advanced Energy, Sun Yat-Sen University (Shenzhen), Shenzhen, Guangdong 518107, PR China. [9]Petroleum and Chemical Industry Key Laboratory of Organic Electrochemical Synthesis, College of Chemical Engineering, and Zhejiang Moganshan Carbon Neutral Innovation Institute, Zhejiang University of Technology, 18 Chaowang Road, Gongshu District, Hangzhou 310032, PR China. [10]Zhejiang Carbon Neutral Innovation Institute, Moganshan Institute ZJUT, Kangqian District, Deqing 313200, PR China. [11]These authors contributed equally: Qilong Wu, Haiyuan Zou, Xin Mao. ✉e-mail: jiayi@zjut.edu.cn; junc@uow.edu.au; yaoxd3@mail.sysu.edu.cn

significantly simplifying the production and transportation scenarios and lowering the manufacturing cost and risks[2]. An alternative route specified for on-site selective electrocatalytic synthesis of $H_2O_2$ via the two electronic pathway of oxygen reduction reaction (2e− ORR) has attracted considerable attention due to its capability to enable the multi-scale reliable production of $H_2O_2$ under ambient conditions[5–8]. Additionally, the cost and environmental footprint of $H_2O_2$ production can be further reduced by coupling it with a renewable energy-driven system[9].

The key to achieving this target is to rationally design highly electroactive catalysts with excellent selectivity for 2e− ORR. To date, various metal-based electrocatalysts have been reported for catalytic $H_2O_2$ production, including noble metals and their alloys (such as, Pd-Au[10], Pt-Hg[9], and Pd-Hg[11]), as well as non-precious metal catalysts (Fe, Co, Ni, Mn, etc.)[6,12–17]. However, the scarcity and high price of noble metals and the potential leaching or degradation of metal species severely hinder their large-scale application[18,19]. Recently, carbon-based metal-free electrocatalysts have shown great potential as alternative catalysts for electrocatalytic synthesis of $H_2O_2$ owing to their low-cost, abundance, high tunability, and chemical stability under identical operational conditions[18,20–24]. It has been reported that the asymmetric distribution of the electronic conjugation structure in carbon, modified by oxygen groups (O-groups) or structural defects, is particularly effective for promoting the 2e− ORR activity[8,18,23,25–27]. However, the catalytic contributions of the afore-mentioned two categories of active sites are frequently indistinguishable because of 1) the coexistence of carbon defects and O-groups on carbons; 2) the ambiguous relationship between carbon defect density and O-groups towards ORR performance; 3) the unclear anchor sites of O-groups on the carbons[28–34]. While some works have proposed potential active site via manipulating the types and distribution of O-groups or using model catalysts[8,18,23,35], the potential dynamic structural transformation during electrocatalysis has been overlooked with few reports. In order to understand the underlying cause of the active site identification controversy, it is important to identify the configuration of the real active site for a 2e− ORR: specific O-groups or defects or other possible co-factor combinations. However, dynamic structural analysis of carbon-based metal-free electrocatalysts is challenging, as it generally involves multiple technology barriers, including atomic structural control, interference exclusion, structural dynamic evolution monitoring, and structural analysis during the reaction process.

In this work, we carry out the investigation to reveal the nature of defect sites via the controllable manipulation of the defect density and the types of O-groups on carbons, including the establishment of the correlation between the defect density and $H_2O_2$ Faradaic efficiencies. As a result, the catalyst with the highest defect density, oxygen modified defective graphene-30 (O-DG-30), shows excellent Faradaic efficiency for $H_2O_2$ production (~98.38 ± 1.6%). A cyclic voltammetry (CV) dependent O-groups redistribution phenomenon has been observed, indicating a dynamic structural transformation of catalysts to further narrow the scope of potential configurations of the active sites. Combined with multiple in-situ technologies and theoretical simulations, including in-situ ATR-IR and Raman spectra, the dynamic evolution processes and kinetic behaviors of intermediates identify that the carbonyl modified pentagon (C5 = O) are the key active sites of the catalyst and the OOH* species are major intermediates of the reaction. The related density functional theory (DFT) calculations also confirm the superiority of the C5 = O structure towards the 2e− ORR, in comparison with other configurations. This work not only reported the dynamic evolution of the active centers but also resolved the topological configuration of active site on defective carbon-based electrocatalysts, providing a potential research paradigm for active site identification in non-model carbon-based metal-free electrocatalysts.

## Results and discussion

### Controllable synthesis and characterization of oxygen modified defective graphene

Theoretically, the difference of the coordination environment such as the in-plane and edge carbon sites hold great impact on the formation energy of O-groups. In an attempt to regulate the modified types of O-groups, we firstly calculated the formation energy of different O-groups on the edge or in-plane carbon sites (Supplementary Fig. 1). As predicted, the formation energies of the O-groups on the edge sites are overall lower than that of in-plane sites. Inspired by this, a series of defective graphene (DG) with different defect densities were controllably synthesized via Ar plasma treatment, aiming to introduce more edges, and expecting to regulate the types of O-groups. As shown in Fig. 1a and Source Data, an obvious ascending trend of $I_d/I_g$ values with the increased Ar plasma treatment time from 5 min to 30 min and reaches a maximal $I_d/I_g$ value of 1.15. With further extension of the treatment time, the $I_d/I_g$ values would decrease gradually, because the excessive radiation induces fusion of carbon pores[29]. Afterwards, the defective graphene was further oxidized to introduce the O-groups, denoted as oxygen modified defective graphene (O-DG) (Supplementary Fig. 2). The powder X-ray diffraction (PXRD) patterns of DGs and O-DGs were collected, which shows a consistent pattern of characteristic peaks, conducive to excluding the influence of Ar plasma treatment and chemical oxidation on the graphene structure, as shown in Supplementary Fig. 3. Transmission electron microscopy (TEM) images of O-DG-30 with highest defect density demonstrated that the Ar-plasma radiation has etched the carbon structure with appearing more carbon holes via comparing the pristine graphene (PG) and the DG-30 (Fig. 1b, c and Supplementary Fig. 4), also similar with reported works[32,36–38]. Moreover, such holes on graphene could be retained after chemical oxidation (Fig. 1d). The high-angle annular dark-field (HAADF) image and elemental mapping of the O-DG-30 indicated that the O-groups have successfully anchored on the defective graphene (Fig. 1e–g). The atomic-level structural characterization of the O-DG-30 was presented by an Integrated Differential Phase Contrast scanning transmission electron microscopy (iDPC-STEM). In the perfect domain of the O-DG-30 (Fig. 1h), the carbon structure shows a typical hexagon lattice and less oxygen atoms (atoms with bright white color). For comparison, in the defective domain, the edges of graphene have been modified clearly by oxygen species and some pentagon defects on carbon edges could be identified directly (Fig. 1i, j and Supplementary Fig. 5), demonstrating the O-groups could be anchored on carbon defects.

### Evaluation of electrochemical $O_2$ reduction performance

The ORR performance of the DG and O-DG catalysts was evaluated, aiming to clarify the electrocatalytic contribution of defect density and O-groups (Source Data). Firstly, DG catalysts were investigated in a 0.1 M KOH aqueous solution using a rotating ring-disk electrode (RRDE), and the corresponding $H_2O_2$ Faradaic efficiencies were calculated and plotted in Supplementary Fig. 7 (the calibration of reference electrodes as shown in Supplementary Fig. 6). With the increase of defect densities, the 4e− pathway of the ORR on DGs was promoted with higher half-wave potential, resulting in lower $H_2O_2$ yield and electron transfer numbers which are similar to the reported works[29,31,38]. Of note, the dominating ORR pathway of the O-DG catalysts is totally reversed to the 2e− pathway after modifying the O-groups on DG catalysts. This reversion probably because the defects with O-groups modification favors the dissociative reduction of *OOH intermediates, in contrary to non-modified defect sites[30]. As shown in Fig. 2a, the linear sweep voltammetry (LSV) curves of ring current ($i_r$) are gradually improved with the increase of the defect density of O-DG samples, coincident with the tendency of $I_d/I_g$ values. Meanwhile, the $H_2O_2$ Faradaic efficiencies were also increased and reached the peak at O-DG-30 as compared to reference samples (52.94% for O-PG, 56.83%

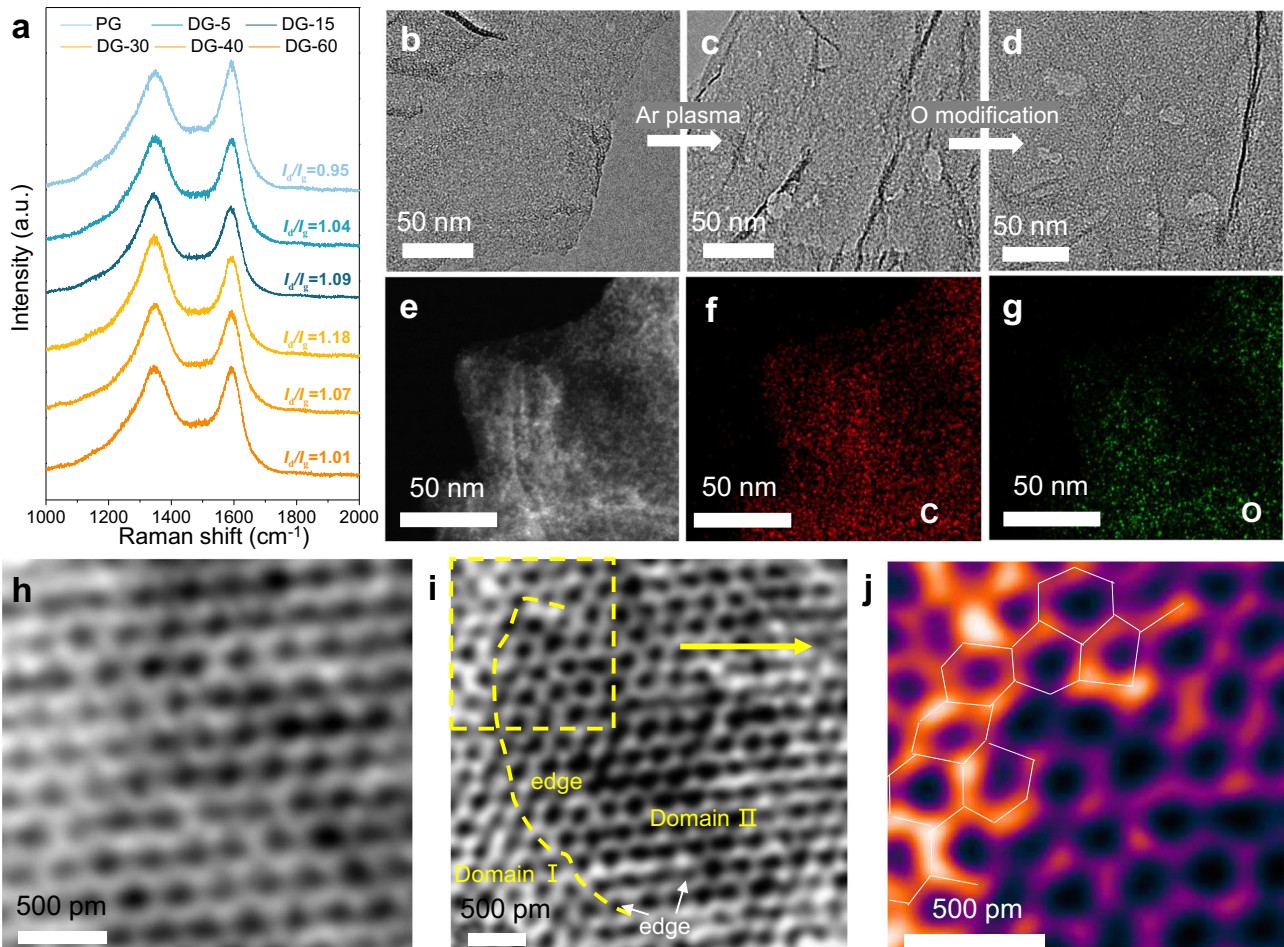

**Fig. 1 | Synthesis and structure characterization of DG and O-DG catalysts.** **a** Raman spectra of pristine graphene and defective graphene (with different Ar plasma treatment time). TEM images of (**b**) PG and (**c**) DG-30 (**d**) O-DG-30. **e** High-angle annular dark-field (HAADF) image of O-DG-30. **f, g** Elemental mapping of O-DG-30. **h** iDPC-STEM image of O-DG-30 in perfect domain. **i** iDPC-STEM image of O-DG-30 in defective domain. **j** Coloring and magnifying iDPC-STEM image of O-DG-30 in defective domain.

for O-DG-5, 62.87% for O-DG-15 and 84.93% for O-DG-30 at 0.55 V vs. RHE), and then decreased with the longer time of Ar plasma treatment (Fig. 2b and Supplementary Figs. 8 and 9). This tendency obviously reveals the strong dependency of 2e⁻ ORR performance of O-DG catalysts on carbon defect density. To understand the effect of O-groups towards 2e⁻ ORR performance, we further compare the $H_2O_2$ Faradaic efficiencies and electron transfer numbers of PG, O-PG, DG-30 and O-DG-30 (Fig. 2c). It shows that the DG-30/O-DG-30 group shows a more obvious improvement of $H_2O_2$ Faradaic efficiency, compared with the non-defect introduced PG/O-PG group, demonstrating that the combination of O-groups and carbon defects would be the real active sites of O-DG catalysts.

Given the low solubility of $O_2$ in the electrolyte, the partial current density of $H_2O_2$ is too low in the RRDE test system to be applied in $H_2O_2$ direct synthesis[39–42]. To this end, a flow cell with three-phase interface was employed to systematically evaluate the ORR performance of O-DG-30 (Fig. 2d and Supplementary Fig. 10a). As shown by the LSV curves in Fig. 2e, the O-DG-30 shows an excellent ORR performance with a low onset potential of 0.9 V vs. RHE and a high current density of 76.6 mA cm⁻² at 0.2 V vs. RHE in the $O_2$-saturated 0.1 M KOH, in contrast to the LSV curve in the Ar-saturated condition. The $H_2O_2$ Faradaic efficiencies of O-DG-30 in flow cell work conditions were measured by UV spectrophotometry (details in the method section and Supplementary Figs. 10b and 11). The Faradaic efficiency of O-DG-30 reaches up to 98.38% at the potential of 0.5 V vs. RHE, comparable with the

state-of-the-art 2e⁻ ORR catalysts (Fig. 2f and Supplementary Table 4)[41]. Of note, the $H_2O_2$ partial current density of O-DG-30 at 0.2 V (vs. RHE) is 71.98 mA cm⁻², leading to a high $H_2O_2$ production rate of 41.45 mg cm⁻² h⁻¹ (Fig. 2g). The long-term stability of O-DG-30 was also investigated at 0.5 V vs. RHE for intermittent and continuous work in the flow cell. In the intermittent working mode, the O-DG-30 catalyst maintained a constant Faradaic efficiency of ~95% after 8 test cycles (Supplementary Figs. 12 and 13). Meanwhile, ~92% of the Faradaic efficiency of the O-DG-30 catalyst in 10 h continuous working mode was retained, further suggesting its excellent stability (Supplementary Fig. 14).

Although the origin of 2e⁻ ORR activity of O-DG-30 has been speculated as the synergistic effect of carbon defect and O-groups, the detailed configuration of active site, especially for the type of active O-groups, is still intangible. In an attempt to identify the actual active O-groups of O-DG-30 catalyst, we preliminarily investigated the types of O-groups on synthesized samples with combination of Fourier Transform Infra-Red (FTIR), X-ray absorption near edge structure (XANES) and X-ray photoelectron spectroscopy (XPS) (Source Data). FTIR was firstly used to identify the difference of the O-groups in the prepared samples. The main O-groups in the GO were assigned to ketones (C=O, ~1750~1850 cm⁻¹), carboxyl (COOH ~1600~1750 cm⁻¹), sp²-hybridized C=C (in-plane stretching, ~1500~1600 cm⁻¹), and epoxides (C-O-C, ~1280~1330 and 800~900 cm⁻¹)[43]. As shown in Fig. 3a, O-PG shows an obvious characteristic peak of C-O-C, in contrast

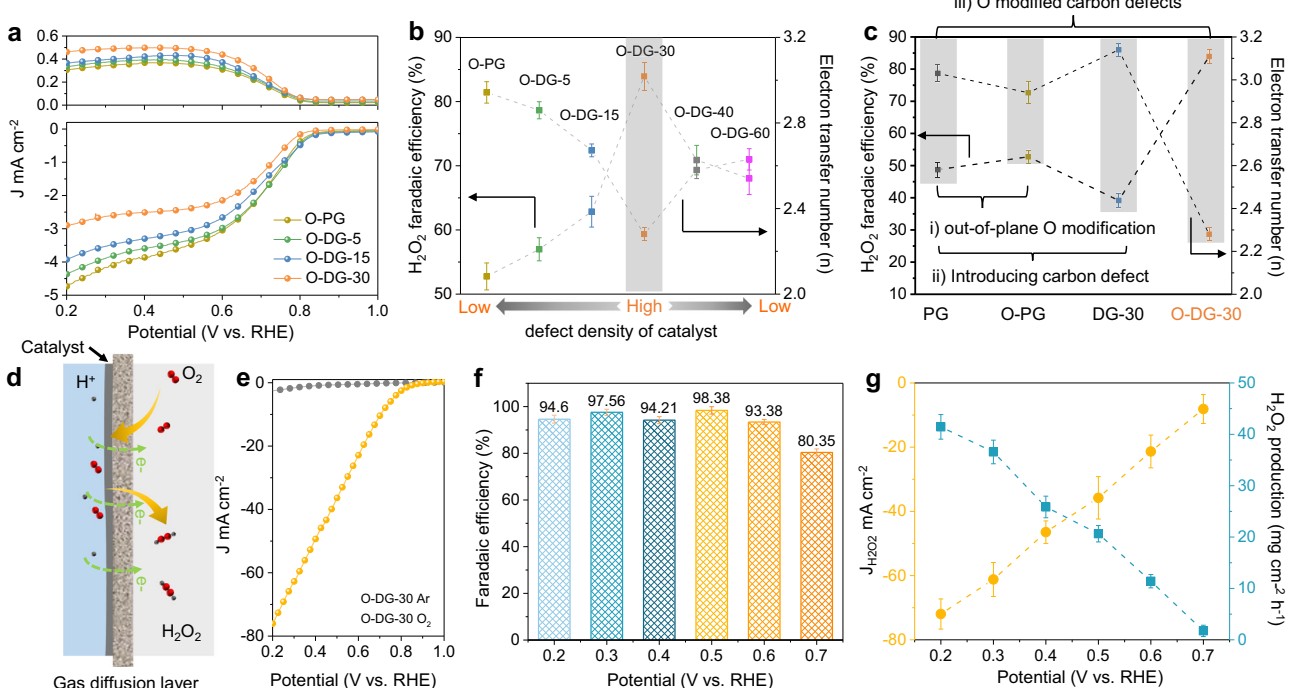

**Fig. 2 | ORR performance of oxygen modified defective graphene catalysts.**
**a** The polarization curves of O-PG, O-DG-5, O-DG-15 and O-DG-30 on RRDE at 1600 rpm in $O_2$-saturated 0.1 M KOH (scan rate: 10 mV s$^{-1}$, resistance in RRDE test system: 21.3 ± 2.4, iR compensation rate: 95%). **b** The $H_2O_2$ faradaic efficiencies of O-PG, O-DG-5, O-DG-15 and O-DG-30 at 0.55 V vs. RHE (The error bars represent two independent samples). **c** The $H_2O_2$ faradaic efficiencies of PG, O-PG, DG-30 and O-DG-30 at 0.55 V vs. RHE. **d** The Schematic diagram of flow cell for $H_2O_2$ production. **e** The

polarization curves of O-DG-30 on flow cell in Ar or $O_2$-saturated 0.1 M KOH (resistance in flow cell test system: 7.2 ± 1.6, iR compensation rate: 95%). **f** The $H_2O_2$ faradaic efficiencies of O-DG-30 on flow cell at different potentials (The error bars represent two independent samples). **g** The partial $H_2O_2$ current density and $H_2O_2$ production of O-DG-30 at different potentials (The error bars represent two independent samples).

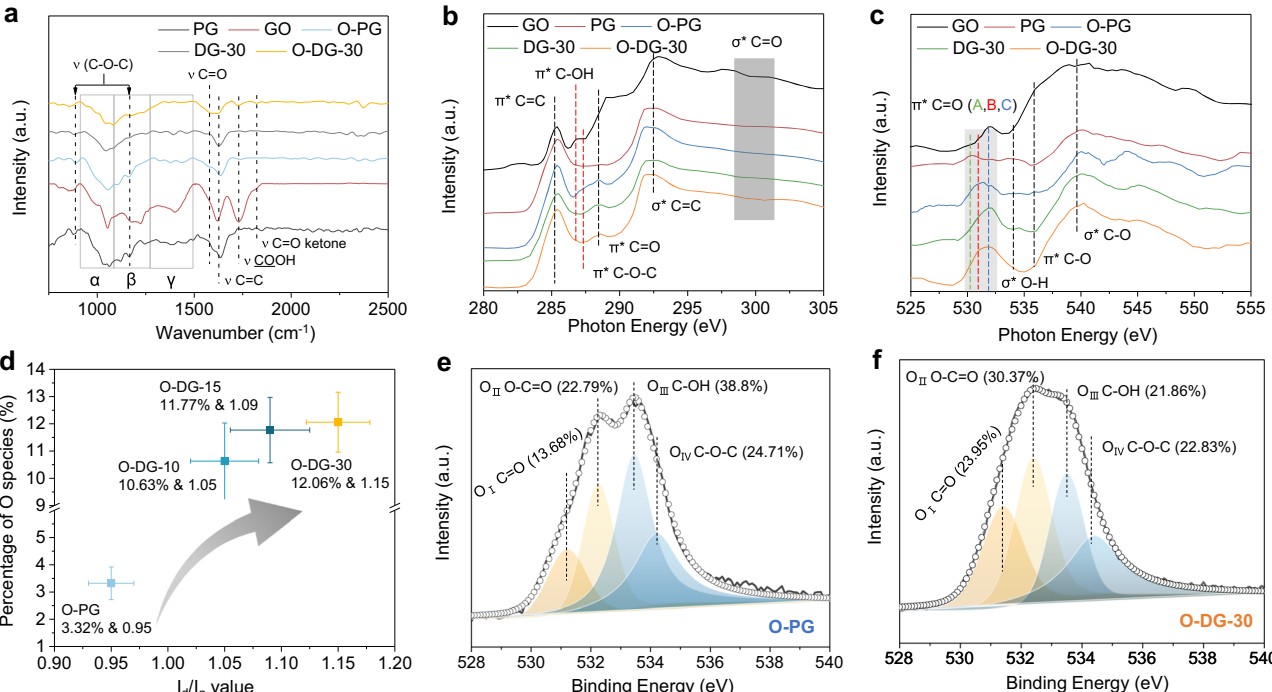

**Fig. 3 | Controllable synthesis of oxygen groups on O-DG catalysts. a** Fourier transform infrared (FTIR) spectra of PG, GO, O-PG, DG-30 and O-DG-30. **b** C K-edge of GO, PG, O-PG and O-DG-30. **c** O K-edge of GO, PG, O-PG and O-DG-30. **d** The

relationship of $I_d/I_g$ values and percentage of O species among O-PG, O-DG-5, O-DG-15 and O-DG-30 (The error bars represent two independent samples). XPS O $1s$ spectra of (**e**) O-PG and (**f**) O-DG-30.

to the O-DG-30 sample, which enabled us to exclude the contribution of the C-O-C group toward the 2e⁻ ORR performance for the O-DG-30. The C=O peak at ~1580 cm⁻¹, which overlaps with the sp²-hybridized C=C in-plane stretching mode, corresponding to the ketones, 1,3-benzoquinones and benzo[de]chromene-2,3-diones[43,44]. Compared with O-PG and DG-30, the signal enhancement of O-DG-30 at peak 1580 cm⁻¹ demonstrated that the chemical oxidation process would produce more C=O groups other than the C-O-C group. Moreover, the O-DG-30 also exhibits a more distinct signal at peaks of 1725 cm⁻¹ and 1820 cm⁻¹ which could be ascribed to carboxyl (COOH) and cyclic ketones (C=O). These results clearly illustrate that the defect sites on graphene are useful for controlling the types of modified O-groups, especially for the O-groups associated with C=O signals.

The soft XANES with high resolution provide more information on bonding configurations. The C K-edge includes unoccupied π* and excited σ* states (Fig. 3b). The 1s–π* C=C peak at 285.3 eV originates from either protonated/alkylated or carbon substituted aromatic-C and is herein referred to aromatic-C[45]. The peak at 286.8 eV can be assigned to π* C-OH since the phenol, phthalic acid, and pyrocatechol are present in the π* C-OH states at 286.5 eV[46]. The peak at 287.5 eV resulted from the charge transfer induced by the O in the etheric ring (π* C-O-C)[23]. As shown, both O-PG and O-DG-30 show obvious π* C=C, C=O signals. However, compared with the absence of peaks at 286.8 (π* C-OH) and 287.5 eV (π* C-O-C) for O-DG-30, O-PG exhibits two minor shoulders in such areas. The results of the C K-edge demonstrated that the catalytic activity of O-DG-30 does not originate from the C-O-C or O-H group but from the C=O group. The information about the O-groups was also investigated by soft O K-edge XANES (Fig. 3c). The broad peak in the energy range of 529.9–533.5 eV indicates the contribution from multiple functional groups of π* C=O, including benzoquinone at 529.9 eV (peak A), ketone at 531.0 eV (peak B) and carboxyl at 531.9 eV (peak C)[45]. As shown, O-DG-30 exhibits much stronger signals at peak A and B, compared with DG-30, indicating that the possible active site could be narrowed to quinone and ketone. Moreover, the absence of σ* O-H and π* C-O signal in the O K-edge of O-DG-30 is conductive to exclude the contribution of hydroxyl and epoxide for the 2e⁻ ORR, in line with the C k-edge results.

To quantify the O-group contents and ratio variation with different defect densities of O-DG samples, XPS was employed and analyzed meticulously. The four samples, O-PG, O-DG-5, O-DG-10, and O-DG-30 were selected as a group because the defect density of them grows gradually, as proved by the Raman spectra. As shown in Fig. 3d and Supplementary Fig. 15, there is a positive correlation between O-group content and defect density. Specifically, under the same chemical oxidation condition, the content of O-groups on the synthesized O-DG samples is remarkably higher than that of O-PG samples, demonstrating that the introduced carbon defects are the anchor sites of O-groups. To further clarify the impacts of carbon defects on the types of introduced O-groups, the high-resolution O 1s spectra of samples were collected. There are four distinct regions that could be classified, which are $O_I$ (531.2 eV, C=O related groups), $O_{II}$ (532.3 eV, COOH), $O_{III}$ (533.4 eV, C-OH), and $O_{IV}$ (534.2 eV, C-O-C), respectively. It is clear that the O-DG-30 exhibits a bigger $O_I$ and $O_{II}$ region but a smaller $O_{III}$ and $O_{IV}$ region, in contrast to the O-PG (Fig. 3e, f). This result indicates that the types of introduced O-groups are greatly affected by the carbon defects in the graphene. The variation trend and relationship between O group types and carbon defect density was also summarized in Supplementary Table 1 (Supplementary Fig. 16). Moreover, the proportion of C=O and COOH regions shows a positive correlation between defect density and $H_2O_2$ Faradaic efficiency (Supplementary Table 1, Figs. 1a and 2b), in contrast to the C-OH and C-O-C regions, which further verifies the actual active site for 2e⁻ ORR originating from C=O and COOH rather than C-OH and C-O-C.

Nevertheless, the contribution of C=O and COOH to the 2e⁻ ORR was still not distinguished. In consideration of the measurement

condition of ORR would intrigue a potential dynamic evolution of O-groups, XPS O 1s spectra of O-DG-30 with different cyclic voltammetry (CV) cycle activations were prudentially studied (Source Data). The XPS was firstly employed to analyze the CV-dependent C/O ratios of O-DG-30, which showed a stable content of O-groups during electrocatalysis (Supplementary Table 2). The distribution and content of O-DGs at different CV cycles are shown in Fig. 4a and Supplementary Table 3. The brown peaks at 535.8 eV are the signal of S-O of Nafion, which could be neglected. As shown, with the increase of CV cycles, the concentration of C=O has increased continuously from 23.95% to 43.86% over 10 CV cycles, while the COOH is increased from 30.37% to 39.31% after the first CV but gradually decrease to 27.20% at subsequent CV cycles. Both C-OH and C-O-C show similar variation tendencies, which decreased after the first CV and were maintained constantly at subsequent CV cycles. The heavy atom replacement experiment was carried via the $NH_3$/Ar-plasma method to exclude the interference of potential N impurities[47]. As shown (Supplementary Fig. 17a), the XRD pattern of $NH_2$-DG-30 shows a consistent pattern of characteristic peaks with O-DG-30. The Raman spectra demonstrated the successful synthesis of defect sites on graphene via $NH_3$/Ar-plasma method, which shows an increased $I_d/I_g$ value compared with pristine graphene ($I_d/I_g = 0.95$) (Supplementary Fig. 17b). Further, the FTIR and XPS spectra were used to investigate the surface N related containing groups of $NH_2$-DG-30 (Supplementary Fig. 17c–e). The broad peaks at 1570 cm⁻¹ can be assigned to the symmetrical stretching vibration and the plane bending vibration of -$NH_2$ groups, which are close to the signal of C=O (-1580 cm⁻¹). The peak at 1159 cm⁻¹ can be attributed to the C-N stretching vibration of amino groups. In addition, the new peak at 1205 cm⁻¹ could be assigned to the C-N vibration of -C-NH-C groups. The XPS survey indicated that around 1.62% N species were introduced to the $NH_2$-DG-30. Deconvolution of the N 1s spectrum of $NH_2$-DG-30 shows three component peaks with the binding energies of 398.8, 399.8 and 401.9 eV, which verify the formation of amide (398.8 eV), amine (399.8 eV), and N⁺ species (401.9 eV) on the graphene substrate, respectively. These results observed in FTIR and XPS have demonstrated that the $NH_2$ was successfully modified on the defective graphene. The absence of $NH_2$ characteristic signals in the FTIR and XPS spectra of O-DG-30 helps to exclude the interference of N related impurities. These results not only reveal a dynamic structural transformation and redistribution of O-groups during electrocatalysis but also identify the real active species (C=O) with possible configurations (pentagon=O) of the O-DG-30 catalyst.

To fundamentally understand the 2e⁻ ORR mechanisms of O-DG-30 catalysts, in-situ attenuated total reflectance infrared (ATR-IR) spectrometry was employed to track the dynamic evolution processes of the relevant surface species during the ORR process (Supplementary Fig. 18a and Source Data). The ATR-IR spectra with different CV cycles represent the change of initial and final states, whereas the ATR-IR spectra at different potentials suggest the dynamic process of catalysis. Therefore, we aim to identify the real active site and relevant surface species during the ORR process via comparing the variation of vibration band positions at different CV cycles and different applied potentials. As shown in Fig. 4c, these two in-situ spectra exhibit similar O-groups signals, including the aromatic C=C group (-1554 cm⁻¹), carbonyl C=O group (-1645 cm⁻¹), carboxyl group (-1745 cm⁻¹) and cyclic ketone group (-1822 cm⁻¹). Meanwhile, the peaks at 1084 cm⁻¹ and 1404 cm⁻¹ were also observed in Fig. 4c and Supplementary Fig. 18b, which could be assigned to OOH* species, $O_{2,ad}$ species according to previous works[18,48]. Notably, the Supplementary Fig. 18a exhibited a slight peak assigned to the $O_2^{-}$* species while it was absent in Fig. 4c, indicating that the OOH* species are the major intermediates of the ORR process of O-DG-30. Moreover, it is worth noting that the peak at 1692 cm⁻¹ in Fig. 4c is much stronger than that in Supplementary Fig. 18b providing significant evidence for further revealing the configuration of the active site and relevant surface species. To further

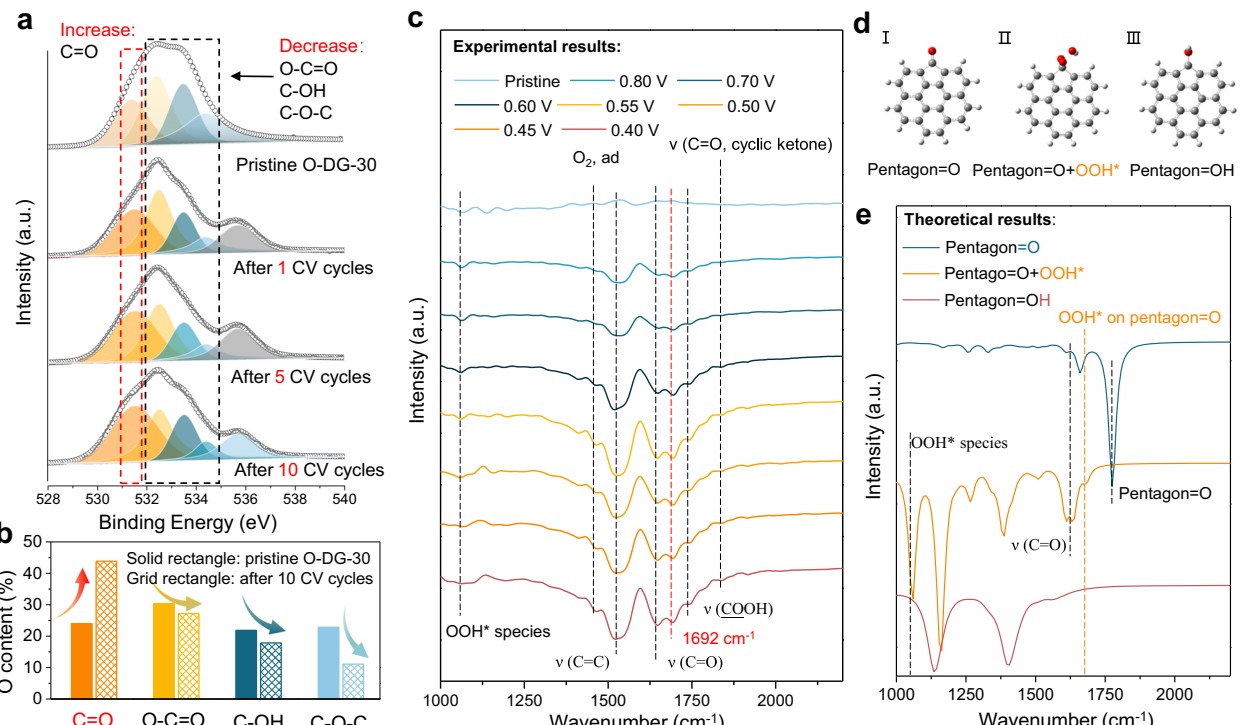

**Fig. 4 | Dynamic evolution monitoring of oxygen groups on O-DG catalysts.**
**a** Ex-situ XPS O *1s* spectra of O-DG-30 with CV activation. **b** The variation of O containing groups after CV activation. **c** In-situ attenuated total reflectance infrared (ATR-IR) spectra of O-DG-30 at various potentials. **d** The possible atomic structures of the O-groups and relevant surface species on defective graphene (Gray: C atom, red: O atom, gray white: H atom). **e** The calculated IR spectra of three possible atomic structures of the O-groups and relevant surface species on defective graphene.

resolve this finding, the theoretical vibration band positions of the O-groups and OOH* species on defective graphene were predicted by density functional theory (DFT) calculations. The predictions of models (Pentagon=O, pentagon=O + OOH* and pentagon=OH) are mainly based on the iDPC-STEM images and in-situ ATR-IR results (Fig. 4d). As shown in Fig. 4e and Supplementary Fig. 19, the theoretical vibration band positions of pentagon=O + OOH* model matched the observed experimental results well. The theoretical peak at 1060 cm$^{-1}$ is much closer to the experimental value of OOH* species. Moreover, the adjacent two theoretical peak positions at 1636 cm$^{-1}$ and 1683 cm$^{-1}$ are consistent with the carbonyl and the unknown signals (~1692 cm$^{-1}$) presented in Fig. 4c. Therefore, the configuration of the active site and relevant intermediates of O-DG-30 during the ORR process could be preliminarily identified.

In order to exclude the contingency of ATR-IR results and further probe any transient adsorbate-catalyst interactions that occurred in the ORR process, we employed an in-situ Raman spectro-electrochemistry cell with a gas-solid-liquid interface to magnify the signal of relevant surface species (Supplementary Fig. 20, Source Data). Both the DG-30 and O-DG-30 electrodes were characterized at various applied potentials in the O$_2$-saturated 0.1 M KOH electrolyte. Of note, no changes in the Raman spectra of DG-30 were observed at these different potentials (Fig. 5a), indicating the stability of the carbon defect during catalysis. For comparison, the O-DG-30 was tested under the same conditions to investigate the O-groups involved electro-catalytic process and interaction with relevant surface species (Fig. 5b). As the applied potential decreased from 0.75 to 0.45 V, noticeable changes in the spectra were detected. At low applied potentials (lower than 0.55 V), several new peaks were observed: such as peakI at 1160 cm$^{-1}$, peakII at 1395 cm$^{-1}$ and peak III at 1537 cm$^{-1}$. Meanwhile, the two major peaks of G and D bands remained unchanged in the 0.75–0.45 V region, which further indicated that the production of new

peaks of O-DG-30 is originated from O-groups that were involved in an electrocatalytic process rather than any change in carbon structure.

To identify the possible atomic structures causing these three peaks, DFT calculation was employed to simulate the possible con-figuration of active sites and relevant surface species, meanwhile compared with in-situ ATR-IR results. The predictions of models (Pentagon=O, pentagon=O + OOH* and pentagon=OH) are the same as the ATR-IR part. As shown in Fig. 5c and Supplementary Fig. 21, the peak depicts the first theoretical vibration band positions (1040 cm$^{-1}$) of pentagon=O + OOH* model well, which also coincides with the OOH* vibration band positions of reported works. The theoretical vibration band position at 1387 cm$^{-1}$ is the D band signal of the pen-tagon=O + OOH* model, which is close to the peak II. Therefore, by combining the ex-situ XPS and in-situ ATR-IR results, the OOH* species attached to the carbonyl group modified carbon defect site (penta-gon=O) could be identified as the key configuration of O-DG-30 during the 2e$^-$ ORR process. Meanwhile, the G band of the pentagon=O + OOH* model appeared at higher vibration band positions (1622 cm$^{-1}$), compared with the pentagon=O model (1580 cm$^{-1}$). This red shift of D and G band peaks of this model may originate from the charge transfer between the attached OOH* species and the pentagon=O site. Con-sidering the surface proton migration during the ORR process, a pentagon=OH model also be simulated. As shown in Fig. 5c, the G band pentagon=OH model is splitted into two parts and the vibration band positions of left peak (1547 cm$^{-1}$) is close to the peak III (1537 cm$^{-1}$). Based on these results, hereby we propose a possible catalytic mechanism of O-DG-30 for 2e$^-$ ORR (Fig. 5d): 1) formation of the pentagon=OH + OOH* structure via the first proton-electron transfer step (FPET); 2) H$_2$O$_2$ formation and release via the second hydro-genation step. The calculated net barrier of H transfer from pentagon=OH + O$_2$* to pentagon=O + OOH* is as low as 0.05 eV (Supplementary Fig. 22), thereby the indirect route from pentagon=O

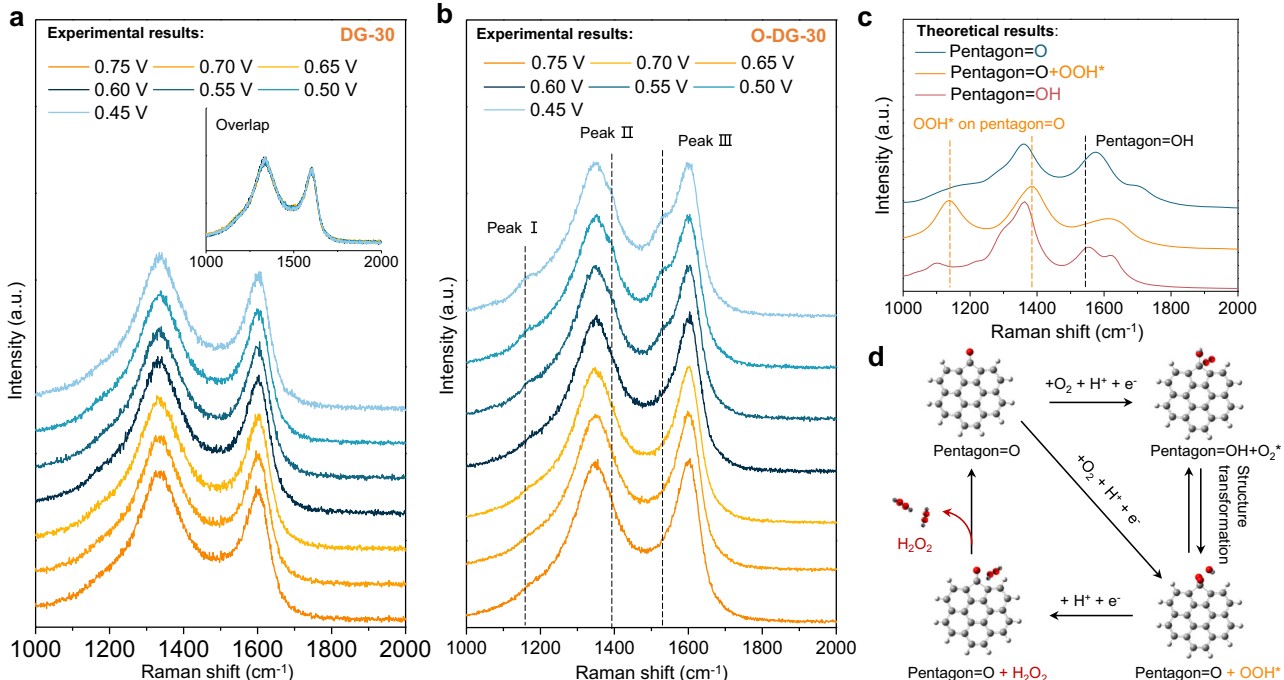

**Fig. 5 | Dynamic evolution monitoring of oxygen groups on O-DG catalysts using In-situ Raman spectroscopy. a** In-situ Raman spectra of DG-30 on flow cell in $O_2$-saturated 0.1 M KOH (the inset figure is the overlap curves of DG-30 spectra at various potentials). **b** In-situ Raman spectra of O-DG-30 on flow cell in $O_2$-saturated 0.1 M KOH (three new peaks were observed with the decrease of applied potentials). **c** The calculated Raman spectra of three possible atomic structures of the O-groups and relevant surface species on defective graphene. **d** Schematic diagram of possible electrocatalytic mechanism of O-DG-30 (Gray: C atom, red: O atom, gray white: H atom).

to pentagon=OH + $O_2^-$* and then to pentagon=O + OOH* is also reasonable, which explains the origin of the peak III.

Although the experimental results have identified the main O-groups, defect configuration and key intermediates, the structure-activity relationship between possible active sites and ORR performance is unestablished. To gain atomic insights about the nature of the active carbonyl modified pentagon defect, we next used density functional theory (DFT) calculations. In view of the coexistence of quinone and carbonyl on O-DG-30, a variety of model structures, mainly including the configurations established by potential O-groups anchored on the defect site located at the zigzag edge and armchair edge, were proposed (Fig. 6a, b, Source Data). In line with previous work, the calculated limiting potential ($U_L$) was employed as the indicator of activity towards $2e^-$ ORR, which is defined as the maximum potential at which the two hydrogenation steps are downhill in free energy (Supplementary Figs. 23 and 24)[9,19].

The calculated results of various configurations are summarized along with the previous work in an activity volcano plot in Fig. 6c. For the $2e^-$ ORR, the thermodynamic equilibrium potential ($U^0 = 0.70$ V) corresponds to the vertices of the activity volcano graphic. Theoretically, a catalyst with the maximum activity should have a $G_{OOH*}$ of 4.22 (0.1) eV. As shown in Fig. 6c, the C1 and C2 models exhibit the lowest activity, suggesting that the carboxyl groups are not the main contributing factors to the ORR activity of O-DG-30, which corresponds to our experimental results well. The D2 model shows the highest activity with a $U_L$ and $\Delta G$ OOH* value of 0.683 V and 4.217 eV, rather very close to the theoretical ideal value. The positions of the B1 and A1 models on the volcano plot are quite near to the D2 model. However, the D model group (D1 + D2 model) displays higher activity than other models (A, B and C model groups) on the whole due to the low activity of B2 and A2 models. Moreover, it is worth noting that the O-groups anchor on the hexagon located at the junction of the zigzag and armchair edge, which is not the enriched site in the carbon lattice. Taking the distribution probability of A and B models during synthesis into

consideration, the A2 and B2 would be the major anchor sites for quinonyl and carbonyl groups, which are much lower than that of the D model group (D1 + D2 model). Therefore, the carbonyl modified pentagon defects (C5=O) could be regarded as the major active sites of O-DG-30 toward $H_2O_2$ production. As a result, the O-DG-30 catalyst, with the highest defect density, provides sufficient sites for specific O-group modification, which in turn serve as highly active sites to maximize the $2e^-$ ORR performance.

In summary, we have developed the design of an O-modified defective graphene (O-DG) catalyst with excellent $H_2O_2$ electrosynthesis activity by identifying the active site with a critical structural configuration and optimizing defect density. The positive correlation between the carbon defect density and the $H_2O_2$ Faradaic efficiency indicates that the combination and specific configuration of defects and O-groups is the key for ORR. Further ex-situ XPS results revealed the redistribution of O-groups during electrocatalytic activation, identified the carbonyl groups as the main active species in the O-DG catalysts, and underlined that the dynamic structural transformation should not be ignored when investigating the catalytic mechanism of carbon-based catalysts. Meanwhile, the detailed configuration of the active sites and reaction intermediates were revealed by combining in-situ ATR-IR, Raman spectra, and DFT spectrum simulations, in-depth understand the catalytic mechanism of the O-DG-30 catalyst. We believe that identifying the active species and the key structural factors through the combination of multiple technologies, especially for the dynamic catalytic process monitoring, will be of great assistance toward the in-depth understanding and rationale designing of carbon-based metal-free electrocatalysts.

## Methods
### Chemicals
Hydrogen peroxide (30%) were purchased from Aldrich, KOH (reagent grade, 90%, flakes) was purchase from Sigma-Aldrich and Graphene was purchased from Xianfeng Nano (XFNANO).

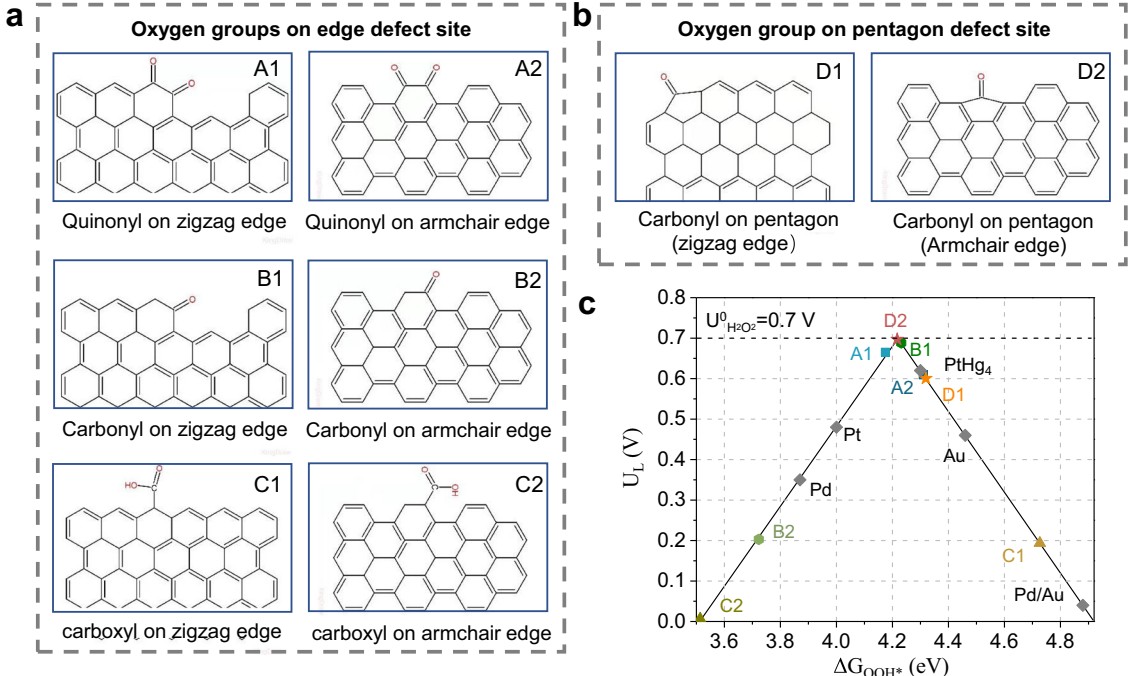

**Fig. 6 | DFT calculations of the 2e⁻ ORR pathway on different oxygen groups modified carbon defect sites. a, b** The atomic structures of the examined O-groups on edge and pentagon defect sites. **c** Theoretical activity volcano plot.

Horizontal dashed line corresponds to the thermodynamic equilibrium potential for 2e⁻ ORR ($U^0_{H2O2}$ = 0.70 V). The activity of alloys are adapted from refs. 9,19, respectively.

## Synthesis of defective graphene (DG)

Defective graphene was fabricated from a commercial graphene powder by Ar plasma treatment under an atmosphere of argon at 100 W. The defect density of defective graphene was controlled by adjusting the time of Ar plasma treatment (form 5 min to 60 min), named as DG-xx (xx means the Ar plasma treatment time).

## Synthesis of oxygen modified pristine graphene (O-PG) and oxygen modified defective graphene (O-DG)

Typically, 60 mg DG was dispersed in 29 mL deionized water for 10 min ultrasound. Then, 1 mL 30% $H_2O_2$ aqueous solution was added and then heated at 100 °C for 2 h under stirring. The as-prepared O-DG (with different Ar plasma treatment time) was purified by centrifuging and washing the above mixture to remove the residual $H_2O_2$ and then dried in vacuum at 60 °C for 6 h. Control group O-PG was prepared by the same method via oxidating graphene directly.

## Materials characterizations

**X-ray diffraction (XRD).** Similar with previous work[49], XRD patterns were recorded on a Rigaku Smart Lab instrument calibrated with Si standard and equipped with Cu Kα radiation (λ = 0.15418 nm) operating at 45 kV and 20 mA. The measurement had a step size of 0.02° over a range of 2 from 10 to 90 degrees. The acquired information was contrasted with the Powder Diffraction File from the database of the International Center for Diffraction Data.

## X-ray photoelectron spectroscopy (XPS)

Similar with previous work[49], XPS measurements were carried by a PHI 5000 VersaProbe II instrument with base pressures less than $5 \times 10^{-10}$ Torr and a monochromatic Mg/Al Kα (1486.6 eV) source (50 W) (spherical spot size of $200 \times 200 \mu m^2$). Survey spectra were recorded with a 280 eV pass energy (step size: 1 eV, dwell times:100 ms). High-resolution spectra were recorded with a 69 eV pass energy (step size: 1 eV, dwell times:100 ms). The PHI-MultiPak 8.2 C was used to process data. XPS spectra were corrected using the C *1s* peak of 284.6 eV.

## Transmission and scanning transmission electron microscopy (TEM/STEM)

The transmission electron microscope (FEI-Tecnai F30) was used to collect the TEM images. The operation energy is 300 kV and equipped with a Schottky field-emission electron gun, a CCD camera and a Digital Micrograph software. The high-angle annular dark-field STEM (HAADF-STEM) imaging and EDS elemental mapping were collected by transmission electron microscope (FEI Talos) operated at 200 kV. Aberration-corrected HAADF-STEM images were performed by a Titan Cubed Themis G20 equipped with a highly sensitive Super-X energy dispersive X-ray detector system (operated at 80 kV).

## Raman spectroscopy

Raman spectra were recorded on a Horiba LabRam Aramis HR Evolution confocal Raman spectrometer with a 633 nm laser. The measure setting parameters of DG samples as shown as follows: 1. acquire time: 5 s; 2. accumulations: 50; 3. grating: 300 (600 nm); 4. ND filter: 10%; 5. Objective: 100x_LWD.

## In-situ infrared/Raman spectroscopy

The in-situ infrared/Raman spectra were recorded with a resolution of 16 s per spectrum at a spectral resolution of 4 cm⁻¹, and the electrode potential was stepwise shifted from 0.75 to 0.45 V; During the in-situ Raman tests, the Raman spectra were recorded against the applied potential from the range of 0.75 V to 0.45 V versus RHE (λ = 633 nm).

## Electrode fabrication

Electrode ink was prepared by mixing the 3 mg of catalyst, 300 uL of DI water, and 300 uL ethanol solution and 30 uL 5 wt % Nafion solution (D520, DuPont). The ink was sonicated to achieve a homogeneous dispersion. The resulting slurry was then uniformly coated onto a glassy carbon electrode (GCE, d = 4 mm) with a catalyst loading of 285 ug/cm².

## Electrocatalysis evaluation on RRDE test system

Electrochemical measurements were carried out in a three-electrode system at 25 °C using an electrochemical station (CHI 760e) and RRDE-3A set-up (ALS, RRDE ver.3.0), similar with the previous work[50]. A modified glassy carbon electrode (GCE, d = 4 mm) served as a working electrode. An Ag/AgCl (3.5 M KCl) and a graphite rod were used as a reference electrode and a counter electrode, respectively. Polarization curves were recorded at a scan rate of 10 mV s$^{-1}$ in 0.1 M KOH electrolytes. All the potentials were converted into the potential versus the reversible hydrogen electrode (RHE) according to E (RHE) = $E_{Ag/AgCl}$ + $E^{\theta}_{Ag/AgCl}$ + 0.05916*pH. The pH of 0.1 M KOH (pH = 12.98 ± 0.02) electrolyte was tested by pH meter and the pH meter was calibrated using pH buffer solution (pH = 6.86). Of note, the purity of reagent grade KOH is 90% which should be counted when configuring the solution. The resistance of all electrochemical cells was tested by the iR compensation function of CHI 760e workstation. The resistance of All the electrochemistry measurements is represented with iR compensation.

The catalyst ink was prepared by dispersing 5 mg catalyst in 1 ml of ethanol-distilled water (1:1) and 50 μl of 5% Nafion aqueous solution (D520, DuPont) by sonication. Then catalyst ink was dropped onto the glassy carbon disk electrode (GCE, 4.0 mm diameter CH Instruments) and dried under ambient condition (catalyst loading on working electrode: 285 ug cm$^{-2}$). Before test, an Ar/O$_2$ flow was used through the electrolyte in the cell about 30 min to saturate it with Ar /O$_2$. Ar /O$_2$-saturated 0.1 M KOH aqueous solution were used as alkaline and acid electrolyte, respectively, where a graphite rod was used as the counter electrode.

The electrons transfer number of (n) and faradaic efficiency of catalysts were derived from the following Eqs. (1) and (2):

$$n = \frac{4I_d}{I_d + I_r/N} \qquad (1)$$

$$H_2O_2(\%) = 200 \times \frac{I_r/N}{I_d + I_r/N} \times 100\% \qquad (2)$$

where $I_d$: disk current; $I_r$: ring current; N: ring collection efficiency (38.6%).

## Electrocatalysis evaluation on flow cell test system

Electrode ink was prepared by mixing the catalyst (5 mg), 0.5 wt % Nafion/DI water solution (0.8 mL), and 0.5 wt % Nafion/ethanol solution (0.2 mL). The ink was sonicated to achieve a homogeneous dispersion. The resulting slurry was then uniformly coated onto a gas diffusion layer of YLS-29BC with a catalyst loading of 0.5 mg/cm$^2$.

## Electrocatalysis evaluation

The electrocatalytic O$_2$ reduction performance of the prepared catalyst (O-DG-30) was evaluated in a gas diffusion flow cell with a three-electrode setup at room temperature. The working electrode is a catalyst-coated gas diffusion layer, the electrode is one new gas diffusion layer, and the reference electrode is Ag/AgCl electrode. An anion exchange membrane (Fumasep FAA-3-PK-130) was used as the compartment separator. The pH of final O$_2$-saturated 0.1 M KOH aqueous electrolyte was 13.25 mL of electrolyte was used in each container. The flow rate of carrier gas O$_2$ was 20 mL/min, and the flow rate of electrolyte was 5 mL/min.

## Product analyses

The production and faradic efficiencies of H$_2$O$_2$ were detected using the method of potassium titanium (IV) oxalate by UV-Vis spectrophotometer at a wavelength of 400 nm. The faradic efficiencies (FEs, %) for H$_2$O$_2$ synthesis was calculated as Eq. (3)

The faradic efficiencies for H$_2$O$_2$ generation were calculated as follows:

$$FE_{H2H2}(\%) = \frac{2CVF}{Q} \times 100\% \qquad (3)$$

where C is the concentration of H$_2$O$_2$ (mol L$^{-1}$), V is the volume of solution (L), F is the Faraday constant (C mol$^{-1}$), and Q is the amount of charge passed through the cathode (C).

## DFT calculations

All the Density functional theory calculations were performed by using the Vienna Ab-initio Simulation Package (VASP) with the Projected Augmented Wave method[51,52]. The exchange-correlation interactions were described by the generalized gradient approximation (GGA)[53] in the form of the Perdew–Burke–Ernzerhof functional (PBE)[54]. For all the geometry optimizations, the cut-off energy was set to 500 eV and the convergence threshold was 10$^{-5}$ eV, and 5 ×10$^{-3}$ eV/Å for energy and force, respectively. At least 15 Å vacuum space was applied in the z-direction of the slab models, preventing the vertical interaction between slabs. The minimum energy pathway for transition states searching was determined by using a climbing image nudged elastic band method (CINEB)[55,56]. For Raman calculation, Gaussian software was used with B3LYP functional and 6-311++G (d,p) base[57]. For the pristine graphene, a 3*7 supercell was built with a unit length of 17.277 Å along x axel, and at least 15 Å vacuum space was applied in the y and z direction of the slab model, preventing interaction between slabs. An automatic k-mesh with Monkhorst-Pack centered 3 × 1 × 1 k points has been carried out to optimize the structure. The formation energies of different oxygen modified defect carbon models were calculated by following formula (4):

$$E_f = E_{total} + m\varepsilon_C - n\varepsilon_O(n\varepsilon_{OH}) - E_{pristine-graphene} \qquad (4)$$

Where, m denotes the number of oxygens, OH spices or COOH group doped in the graphene, and n demotes the number of carbon atom defects. $\varepsilon_C$ and $\varepsilon_O$ are calculated from stable graphene unit cell and oxygen molecules. $E_{total}$ and $E_{pristine-graphene}$ refers to the calculated energy for the deficient graphene and pristine graphene model.

## Data availability

The data that support the plots within this paper and other finding of this study are available from the corresponding author upon reasonable request. Source data are provided with this paper.

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

## Acknowledgements

The support from Ministry of Science and Technology (MOST) of China through the key project of research & development (2021YFF0500502) is appreciated. Q.L. thanks to discovery project of Australian Research Council (DP200101105). Y.J. also thanks Zhejiang Province Overseas High-level Talents Plan (Long-term Innovation).

## Author contributions

Y.J. and X.D.Y. conceived and designed the project. X.D.Y., J. C. and Y.J. supervised the project. Q.W., J.H., L.W. and C.L. prepared the samples and performed the electrocatalytic measurements. Q.W., H.Z., Y.S., S.C., B.Z., and L.S. performed the characterizations, including XRD, Raman, SEM, TEM, iDPC-STEM, FTIR, XAS, XPS, in-situ FTIR and Raman, etc. Q.W., H.Z., X.M. and Y.J. analyzed the synthetic and electrocatalytic mechanism. X.M. and A.D. performed the DFT calculations. Q.W. and Y.J. and X.D.Y wrote the manuscript. X.C.Y., Q.L., J.C and X. W. help to revise the manuscript. Q.W., H.Z. and X.M. contributed equally to this work. All the authors discussed the results and commented on the manuscript.

## Competing interests

The authors declare no competing interests.
