## [Peer Review File · Nature Communications]

Editorial Note: This manuscript has been previously reviewed at another journal that is not operating a transparent peer review scheme. This document only contains reviewer comments and rebuttal letters for versions considered at *Nature Communications* .

REVIEWER COMMENTS

Reviewer #1 (Remarks to the Author):

The additional information provided by the authors clearly demonstrates the low selectivity of the carbon-based materials synthesized in this work. There are already various similar reports in the literature that show much higher selectivity for H₂O₂ synthesis. This work clearly does not meet the high standard of Nature publication.

Reviewer #2 (Remarks to the Author):

Accept, my previous comments and questions are answered adequately.

Reviewer #4 (Remarks to the Author):

The revision didn't provide sufficient improvements to match Nature Communications. After the authors replaced the old Raman spectra with new ones, more concerns were raised about the reliability of the data.

Response letter to reviewers

Reviewer #1: The additional information provided by the authors clearly demonstrates the low selectivity of the carbon-based materials synthesized in this work. There are already various similar reports in the literature that show much higher selectivity for H₂O₂ synthesis. This work clearly does not meet the high standard of Nature publication.

Response: We actually replied this question in our last revised version. As shown in Table S4, it is clearly demonstrated that our catalyst performs better than reported data in literature, in terms of both selectivity and current density.

Reviewer #2: Accept, my previous comments and questions are answered adequately.

Response: Thanks a lot for your careful review and positive comments.

Reviewer #4: The revision didn't provide sufficient improvements to match Nature Communications. After the authors replaced the old Raman spectra with new ones, more concerns were raised about the reliability of the data.

Response: Thanks for your comments. At first, we are sorry for our unclear response last time. We understand the reviewer's concern about the reprehensibility of the Raman data. Herein, we will explain the reasons for the two Raman data deviations in details.

1) Operation procedure and parameter setting of the Raman spectra test: Raman spectra were recorded on a Horiba LabRam Aramis HR Evolution confocal Raman spectrometer with a 633 nm laser (**Figure R1**). The standard operating procedure (SOP) of LabRAM HR Evolution-RAMAN could be found on the official web

(http://www.cen.iitb.ac.in/slotbooking/SOP/350_SOP.pdf). The measure setting parameters of DG samples are shown as follows:

1. acquire time: 5 s
2. accumulations: 50
3. grating: 300 (600 nm)
4. ND filter: 10%
5. Objective: 100x_LWD

Figure R1 The parameters setting procedure of Raman spectrometer.

- 2) Reasons for the different signal-to-noise ratios: As known, the Raman laser with its high output energy can easily burn carbon materials, especially for the defective graphene. At the beginning, for the old Raman spectra test (PG, DG-5, DG-15, DG-

30), we set the ND filter parameter to 5% to avoid the collapse of samples (**Figure R2**). Of note, the higher ND filter parameter enables a smoother Raman curve without other significant impacts on the data if the sample is still stable under this condition. Thus, our cooperater helps us to replenish the Raman data of DG-40 and DG-60 to look for the peak of the I_d/I_g value of defective graphene using a higher ND filter value (10%). We are sorry that we didn't notice the difference in signal-to-noise ratios in the old Raman data. In the new Raman data, we redetected the Raman spectra of PG, DG-5, DG-15 and DG-30 by changing the ND filter parameter to 10%. In this case, the setting parameters of all samples are the same, and the signal-to-noise ratios of the above samples are also reduced.

Figure R2 a) The old Raman data of DG samples. b) The new Raman data of DG samples.

3) A Raman spectra test description was also added in the manuscript (method part).

Horiba LabRam Aramis HR Evolution confocal Raman spectrometer with a 633 nm laser. The measure setting parameters of DG samples as shown as follows: 1.acquire time: 5 s; 2.accumulations: 50; grating: 300 (600 nm); ND filter: 10%; Objective: 100x_LWD.